# Polyamide 6/Poly(vinylidene fluoride) Blend-Based Nanocomposites with Enhanced Rigidity: Selective Localization of Carbon Nanotube and Organoclay

**DOI:** 10.3390/polym12010184

**Published:** 2020-01-10

**Authors:** Hung-Ming Lin, Kartik Behera, Mithilesh Yadav, Fang-Chyou Chiu

**Affiliations:** 1Department of Chemical and Materials Engineering, Chang Gung University, Taoyuan 333, Taiwan; twhmlin@gmail.com (H.-M.L.); b.kartik1991@gmail.com (K.B.); dryadavin@gmail.com (M.Y.); 2Department of Chemistry, Prof. Rajendra Singh Institute of Physical Sciences for Study and Research, V. B. S. Purvanchal University Jaunpur, Uttar Pradesh 222003, India; 3Department of General Dentistry, Chang Gung Memorial Hospital, Taoyuan 333, Taiwan

**Keywords:** polyamide 6, poly(vinylidene fluoride), blend, nanocomposites, physical properties

## Abstract

Polyamide 6 (PA6)/poly(vinylidene fluoride) (PVDF) blend-based nanocomposites were successfully prepared using a twin screw extruder. Carbon nanotube (CNT) and organo-montmorillonite (30B) were used individually and simultaneously as reinforcing nanofillers for the immiscible PA6/PVDF blend. Scanning electron micrographs showed that adding 30B reduced the dispersed domain size of PVDF in the blend, and CNT played a vital role in the formation of a quasi-co-continuous PA6-PVDF morphology. Transmission electron microscopy observation revealed that both fillers were mainly located in the PA6 matrix phase. X-ray diffraction patterns showed that the presence of 30B facilitated the formation of γ-form PA6 crystals in the composites. Differential scanning calorimetry results indicated that the crystallization temperature of PA6 increased after adding CNT into the blend. The inclusion of 30B retarded PA6 nucleation (γ-form crystals growth) upon crystallization. The Young’s and flexural moduli of the blend increased after adding CNT and/or 30B. 30B exhibited higher enhancing efficiency compared with CNT. The composite with 2 phr 30B exhibited 21% higher Young’s modulus than the blend. Measurements of the rheological properties confirmed the development of a pseudo-network structure in the CNT-loaded composites. Double percolation morphology in the PA6/PVDF blend was achieved with the addition of CNT.

## 1. Introduction

Polymer blends filled with nanofillers have attracted considerable attention in academia and in the industry because of their superior properties compared with conventional blends. Polyamide 6 (PA6) is an essential crystalline engineering plastic with good chemical stability, high strength, excellent wear resistance, and oil resistance. The crystal structures in PA6 are mainly α- and γ-morphism [1,2,3,4]. The α-form is composed of a fully extended planar zigzag chain conformation, while the γ-form consists of pleated sheets of parallel chains joined by hydrogen bonds [5]. The polar functional groups (amide) of PA6 tend to absorb moisture, resulting in poor stability after prolonged exposure to air environment. Poly(vinylidene fluoride) (PVDF), another crystalline engineering plastic, possesses high mechanical strength, excellent thermal/chemical stability, and good hydrolysis resistance and weatherability [6,7]. PVDF has five distinct polymorphs (α, β, γ, δ, ε).The polar β-structure-induced piezo- and pyroelectric characteristics of PVDF allows for applications in sensors and actuators [8,9]. Through polymer blending approach, it is expected to overcome the drawbacks of parent components. The PA6/PVDF blends and related composites with fine dielectric properties can be applied in an electronic field, such as embedded capacitors [10]. PA6/PVDF blend-based membranes may be used for separating gases such as CO_2_, N_2_, CH_4_ [5]. Accordingly, PA6/PVDF blends may integrate the advantages of individual components to present broad applications in various fields.

Various combinations of polymers and nanofillers have been used to fabricate high-performance polymer nanocomposites. Chiu et al. [11] reported that adding organically modified montmorillonite clay (OMMT) induced the formation of γ -form PA6 crystals. The inclusion of carbon nanotube (CNT) favored the formation of α-form PA6 crystals in the nanocomposites [5,12]. Pramoda et al. [13] reported that the incorporation of OMMT assisted the formation of β-form PVDF crystals. Chiu [14] differentiated the physical properties of PVDF/OMMT and PVDF/CNT nanocomposites. The addition of OMMT induced β-form PVDF crystal growth, but CNT barely affected the development of the original α-form crystals. Both nanofillers facilitated the nucleation of PVDF, but CNT exhibited a higher nucleation efficiency than OMMT. In addition to the binary polymer/nanofiller nanocomposites, polymer blend-based multicomponent nanocomposites with superior mechanical and thermal properties due to the modification on phase morphology were reported [15,16,17,18].

Huitric et al. [19] reported that adding up to 1 wt % of OMMT to the linear low-density polyethylene/PA12 blend increased the Young’s modulus by 57%. Chiu et al. [20,21] characterized PVDF/poly(methyl methacrylate) blend-based nanocomposites with OMMT and CNT as nanofillers. The storage modulus values of the binary and ternary composites increased to 23% and 54% at 25 °C, respectively, compared with that of PVDF. Li et al. [22] studied PVDF/polycaprolactone (PCL)/CNT nanocomposites and found that CNT was selectively localized in the PCL phase. Mao et al. [10] reported altering the weight ratio of PVDF-to-PA6 resulted in the evolution of the sea-island morphology to co-continuous morphology. CNT was selectively localized in the PA6 phase in both blends with sea-island and co-continuous morphologies because of the finer interaction between PA6 and CNT. Phase transformed from sea-island to co-continuous by CNT loading, which increased the melt viscosity of the PA6 phase [23]. Systems with hybrid filler are recently studied to fabricate high-performance polymer nanocomposites. Chen et al. [24] reported that with the inclusions of CNT and/or OMMT, the blend-based composites of polycarbonate/PVDF showed apparent change in the morphology from a typical sea-island structure to the quasi-co-continuous structure. Chiu [25] studied the effect of loading halloysite nanotube (HNT) and OMMT on the physical properties of poly[(butylene succinate)–*co*–(adipate)] (PBSA)/maleated polyethylene blend-based nanocomposites. The rigidity of PBSA evidently increased after the formation of blend-based nanocomposites. Loading 4 wt % HNT exhibited higher enhancing efficiency for the rigidity of the blend than loading 5 wt % OMMT. 

The influences of individual and simultaneous incorporations of OMMT and CNT on the physical properties of polymer blends merit thorough investigation from the academic research and practical application viewpoints. OMMT- and CNT-loaded PA6/PVDF blend-based nanocomposites have not yet been reported. In the current study, OMMT and CNT were utilized individually and simultaneously for preparing PA6/PVDF blend-based nanocomposites by using a twin screw extruder. This study was performed to investigate the change in the morphology of the immiscible blend after the incorporation of single or hybrid nanofillers. The crystallization/melting behavior and the mechanical and rheological properties of prepared blend and composite samples were compared and reported.

## 2. Experimental 

### 2.1. Materials

PA6 (TP 4208) was a commercial product of Zig Sheng Industrial Co. (Taiwan). PVDF (Kynar® 710), with an average molecular weight of ca. 1.9 × 10^5^ g mol^−1^, was provided by Arkema (France). Organomontomorillonite (Cloisite® 30B, denoted as 30B) was obtained from Southern Clay Products, Inc (USA). Multi-walled CNT, with a carbon purity of >95%, an average outer diameter of 60 nm, and an aspect ratio of >100, was supplied by Golden Innovation Co. (Taiwan). N,N-Dimethylformamide (DMF), which was used for etching experiments, was purchased from JT Baker (Radnor, PA, USA).

### 2.2. Sample Preparation

All samples were prepared by using twin screw extruder (SHJ-20B, L/D = 40) in co-rotating mode. The screw speed was maintained at approximately 200 rpm. The barrel temperatures from the hopper to the die were maintained within 180–240 °C. Before melt-mixing, the ingredients were dried at 70 °C for 24 h in an air-circulated oven to remove the absorbed moisture. After melt-mixing, the test specimens were prepared by injection molding (V4-20SP-G, Multiplas Enterprise Co., Ltd., Taoyuan, Taiwan). PA6 blended with PVDF at weight ratio 50/50 was represented as A5F5. The blend filled with 2 phr of 30B or CNT was denoted as A5F5C2 and A5F5T2, respectively. The blend filled with 1 phr each of 30B and CNT was denoted as A5F5C1T1.

### 2.3. Characterizations

The morphology of the prepared blend and composites was examined by a field emission scanning electron microscope (FESEM, JSM-7500, Hitachi High-Technologies Corp., Tokyo, Japan). FESEM images were observed from cross-section of cryo-fractured specimens (in liquid nitrogen for five min). The SG capture software was used to measure the domain size of PVDF from FESEM images. Transmission electron microscope (TEM, JEM-2000EX II, Jeol Ltd., Akishima, Japan) at an acceleration voltage of 100 kV was employed to assess the dispersibility of 30B and CNT. The TEM samples (ca. <100 nm) were prepared by ultra-cryomicrotoming at −130 °C. X-ray diffraction (XRD) was performed using a Bruker instrument (D2, Billerica, MA, USA) to investigate the crystal structures of PA6 and PVDF in the samples. The X-ray source was CuKα radiation with a wavelength of 0.154 nm. The diffractograms were scanned within 2θ of 10°–30°, and the X-ray unit was operated at 40 kV and 30 mA. 

A differential scanning calorimeter (TA DSC Q10, Milford, MA, USA) equipped with an intercooler under a nitrogen environment was utilized to investigate the crystallization and melting behavior of the samples. For the non-isothermal crystallization experiments, the samples were first heated from room temperature to 240 °C at 20 °C/min, and then cooled to room temperature at different rates (5 and 40 °C/min). The crystallized samples were heated at 20 °C/min to observe the melting behavior. The samples were heated to 240 °C, cooled at a fast rate of 90 °C/min to the predetermined crystallization temperatures (*T*_c_s), held for various times, and subsequently heated to 240 °C (20 °C/min) to study isothermal crystallization and melting behavior. The Young’s modulus (YM), yield strength (YS), and flexural modulus (FM) were measured at a crosshead speed of 1 mm/min, using a Gotech Al-2000 system (Taichung, Taiwan). For YM and YS measurement, the dumbbell-shaped specimen was used according to ASTM D638. For FM measurement, the cuboid-shaped specimen was used (L:W:H = 62.5:12.5:3.2) according to ADTM D790. Six specimens were tested for each formulation, and the average value was reported. The conditions we chose for the different tests were to obtain the proper data for comparison. Rheological properties were determined at 240 °C by using an Anton Paar Physica rheometer (MCR 101, Anton Paar GmbH, Graz, Austria) in oscillating mode with parallel plate geometry (25 mm diameter and 1 mm gap).

## 3. Results and Discussion

### 3.1. Phase Morphology and Selective Localization of Nanofillers

The cryo-fractured surfaces of the representative samples were investigated using FESEM (Figure 1). Figure 1a shows the biphasic morphology of the A5F5 blend, indicating its immiscible characteristics. Based on the difference in the densities of PA6 (1.14 g/cm^3^) and PVDF (1.78 g/cm^3^), the volume ratio of PA6 to PVDF in the blend was 61:39. The PVDF should play the dispersion phase (domains) in the blend/composites. The dispersed PVDF exhibited an average domain size of 3.1 µm and was composed of 17% with size <2 µm, 64% with size 2–4 µm, and 20% with size >4 µm. Figure 1b shows the evidently smaller and more uniform PVDF domain size in A5F5C2 compared to A5F5. This result indicates that 30B played a compatibilizer role to improve the compatibility between PA6 and PVDF. For the composite with 2 phr CNT (Figure 1c), the biphasic structure became diffused and some elongated domains of PVDF were observed. The CNT was mainly dispersed in the PA6 matrix phase, and some CNT particles were found across the PA6-PVDF two phases. CNT played a crucial role in the formation of quasi-co-continuous PA6-PVDF phases. A5F5T2 composite showed evident quasi-co-continuous morphology. With the loading of 30B and CNT into the blend (cf. A5F5C1T1, Figure 1d), the morphology was in between those of A5F5C2 and A5F5T2, and the quasi-co-continuous PA6-PVDF morphology was relatively developed.

The scanning electron micrographs of PVDF-etched (by DMF) A5F5T2 and A5F5C1T1 are shown in Figure 1e,f to determine the component responsible for the dispersion phase in the blend/composites. The dispersed phase was removed (etched out), confirming that PVDF played the dispersed domains. The dispersion of CNT in the PA6 matrix was evident again. The dispersibility of 30B within the composites was examined by TEM. Figure 2a,b show the transmission electron micrographs of the finely dispersed 30B within the PA6 matrix of A5F5C2 and A5F5C1T1. 30B was also detected at the interface between PA6 and PVDF. CNT was also found in the PA6 phase and in the interfacial region of PA6-PVDF phases. The selective localization of 30B and CNT mainly within the PA6 matrix phase was determined. The multicomponent PA6/PVDF blend-based nanocomposites was achieved.

Gomari et al. [26] and Zhao et al. [27] discussed the location of nanofillers in immiscible blends by assessing the wetting coefficient,${\;\omega }_{\alpha }$, according to Young’s equation [26]. For the PA6/PVDF blend system, the equation can be expressed as follows:(1)$$\omega_{\alpha }=\;\frac{\gamma_{\mathrm{PVDF}-\mathrm{filler}}-\gamma_{\mathrm{PA}6-\mathrm{filler}}}{\gamma_{\mathrm{PVDF}-\mathrm{PA}6}}$$
where $\gamma_{\mathrm{PVDF}-\mathrm{filler}}$ is the interfacial energy between PVDF and the nanofiller, $\gamma_{\mathrm{PA}6-\mathrm{filler}}$ is the interfacial energy between PA6 and the nanofiller, and $\gamma_{\mathrm{PVDF}-\mathrm{PA}6}$ is the interfacial energy between PVDF and PA6. In Equation (1), if $\omega_{\alpha }\;$> 1, then the nanofiller will be distributed in PA6 phase. If $\omega_{\alpha }\;$< −1, the nanofiller will be distributed in the PVDF phase. If − 1 <$\;\omega_{\alpha }\;$< 1, the nanofiller will be located at the interface between the two phases. The interfacial energy between the two components could be calculated from the surface tensions of the components by using two different approaches. The first approach is the geometric mean equation [26], which is suitable for polar systems as follows:(2)$$\gamma_{12}=\gamma_{1}+\gamma_{2}-2\left(\sqrt{\gamma_{1}^{d}\gamma_{2}^{d}}+\sqrt{\gamma_{1}^{p}\gamma_{2}^{p}}\right)$$
The second approach is Wu’s harmonic mean equation [26] as follows:(3)$$\gamma_{12}=\gamma_{1}+\gamma_{2}-4\left(\frac{\gamma_{1}^{d}\gamma_{2}^{d}}{\gamma_{1}^{d}+\gamma_{2}^{d}}+\frac{\gamma_{1}^{p}\gamma_{2}^{p}}{\gamma_{1}^{p}+\gamma_{2}^{p}}\right)$$
where $\gamma_{1}$ and $\gamma_{2}$ are the surface tension of components 1 and 2, respectively. $\gamma_{}^{d}$ and $\gamma_{}^{p}$ stand for the dispersive and polar parts of the surface tensions, respectively.

In this study, the surface tensions of PA6, PVDF, and two nanofillers were derived from the literature [27,28,29,30]. The retrieved values were corrected to correspond to the mixing temperature (240 °C) by using the temperature coefficients (−dγdT^−1^) in literature [26]. According to Equations (2) and (3), the interfacial tensions between the pairs of components are summarized in Table 1. By setting the nanofiller as 30B or CNT, the values of $\omega_{\alpha }$ according to the geometric mean and harmonic mean equations were calculated using Equation (1). The results are presented in Table 2. Based on the calculated $\omega_{\alpha }$, 30B or CNT were anticipated to be located in the PA6 phase. The obtained morphology results were mostly consistent with the predicted results. However, some 30B and CNT were observed at the interfacial region of the two phases.

### 3.2. Crystal Structure

Figure 3a shows the XRD patterns of the precooled samples (10 °C/min from the melting state). Neat PA6 exhibited characteristic diffraction peaks of the α-form crystal at 2θ = 20.5° (200) and 23.7° (202/002), whereas neat PVDF revealed characteristic diffraction peaks of the α-form crystal at 2θ = 17.9° (100), 18.5° (020), 20.0° (110), and 26.6° (021) [31,32,33,34]. The blend and CNT-loaded composites showed diffraction peaks corresponding to the individual PA6 and PVDF components. The presence of CNT did not alter the crystal structures of PA6 and PVDF. However, only 30B-loaded composite (A5F5C2) exhibited additional diffraction peaks at approximately 2θ = 11.0° (020) (weak intensity) and 21.5° (200/001), which indicates the formation of the α/γ-crystalline form of PA6. With the presence of 30B and CNT, the diffractions associated with the γ-crystalline form of PA6 was hardly detected. Figure 3b shows the XRD patterns of the air-quenched samples cooled from the melt state to room temperature by air circulation. Neat PA6 showed characteristic diffraction peaks at approximately 2θ = 20.5° and 23.7° (α-crystalline form) and 21.5° (γ-crystalline form). Neat PVDF showed diffractions of the α-crystalline form. The diffraction peaks of the A5F5 blend were superimposed of the individual PA6 and PVDF components. The PVDF in the composites showed α-crystalline form, except with minor diffraction of β-crystalline form in the A5F5C2. The γ-crystalline form of PA6 became more evident in the A5F5C2 composite where only 30B was added. When only CNT was loaded in the composite (cf. A5F5T2), PA6 exhibited mainly α-form crystal diffraction. The A5F5C1T1 samples showed the diffractions of the γ-crystalline form of PA6 and the α-crystalline form of PA6 and PVDF. The XRD results revealed that 30B induced the β-crystalline form of PVDF (under fast cooling process) and the γ-crystalline form of PA6. The addition of CNT did not alter the α-crystalline form of PVDF, but diminished the growth of the 30B-induced γ-crystalline form of PA6. The induction of β-form PVDF crystals with CNT/clay presence has been previously reported [12,35]. In this study, the selective location of 30B and CNT mainly within the PA6 phase was responsible for the lack of change in the crystal form of PVDF in the composites. Thus, crystal form modification was evident for PA6 in the composites.

### 3.3. Crystallization and Melting Behavior

The effects of blending with its counterpart and addition with 30B and/or CNT on the crystallization behavior of PA6 and PVDF were investigated through DSC. Figure 4a shows the 5 °C/min cooling curves of the samples. The crystallization peak temperature (*T*_p_) of the neat PA6 and PVDF were 194.1 °C and 139.0 °C, respectively. After blending, the *T*_p_ of PA6 slightly shifted to a lower value while PVDF exhibited a lower temperature crystallization in addition to the original crystallization peak. The lower temperature crystallization was associated with the small domain-caused homogeneous crystallization (SEM results). For the sole 30B-loaded composite (cf. A5F5C2), the *T*_p_ of PA6 slightly decreased because of the growth of γ-form crystals of PA6 (XRD data). The PVDF showed a *T*_p_ (uniform domain size) similar to that of neat PVDF. With the loading of CNT in the blend (cf. A5F5T2 and A5F5C1T1), the *T*_p_ of PA6 shifted to higher temperatures because of the nucleation effect of CNT. The *T*_p_ of PVDF marginally changed compared with that of the neat PVDF. The quasi-co-continuous PA6-PVDF morphology caused partially overlapped two crystallizations of PVDF in A5F5T2. The determined *T*_p_ and crystallinity values of the samples are listed in Table 3. The crystallinity of PA6 and PVDF of the selected samples are calculated from the melting enthalpy (∆*H*_m_) of the thermograms according to Equation (4) [36]:(4)$$\chi_{c}=\frac{∆Hm}{{w\Delta Hm}^{o}}$$
where *w* is the weight fraction of PA6 or PVDF in the composite, ∆*H*_m_ is the heat of fusion at the melting point, and Δ*H*_m_^o^ is the heat of fusion of 100% crystalline PA6 or PVDF. The estimated Δ*H*_m_^o^ values of PA6 and PVDF are 230.0 J/g [37] and 104.6 J/g [21], respectively. Table 3 shows a decrease in the PVDF crystallinity in the composites, possibly due to the hindrance to the diffusion of PVDF chains. By contrast, the PA6 crystallinity increased in the composites compared with the blend, likely because CNT and 30B were mainly located in the PA6 phase (SEM image). A formulation-dependent crystallization behavior similar to that of the samples cooled at 5 °C/min was observed in the samples cooled at 40 °C/min (Figure 4b). The *T*_p_ shifted to lower values for the fast-cooled samples mainly because of the thermal lag effect. The isothermal crystallization behavior of PA6 in different samples was investigated through DSC. The isothermal crystallization curves of A5F5 and A5F5C2 at various temperatures as a function time *t* are presented in Figure 4c,d. According to the nucleation-controlled crystal growth theory, the crystallization time increased with increasing crystallization temperature (*T*_c_). The plots in Figure 4e depict the reciprocal of the crystallization peak time (*t*_p_) versus *T*_c_. The reciprocal value (*t*_p_^−1^) is proportional to the overall crystallization rate. Neat PA6 and A5F5 showed similar isothermal crystallization rates at individual *T*_c_. The well-dispersed 30B in the PA6 matrix phase (TEM image for A5F5C2 in Figure 2) caused the interaction between PA6 and 30B, and thus induced the slowly crystallized γ-form PA6 crystals. The CNT-loaded composites (c.f. A5F5T2 and A5F5C1T1) showed a faster PA6 crystallization rate than the other samples at individual *T*_c_ because of the CNT nucleation effect. The presence of 30B (induced γ-form crystals) reduced the CNT nucleation effect for PA6 crystallization compared with the A5F5T2 sample at different *T*_c_s.

Figure 5a shows the comparison of the DSC melting behavior of the neat components, blend, and composites precooled at 5 °C/min. Neat PA6 and PVDF exhibited melting temperature (*T*_m_) at 220.9 °C and 172.0 °C, respectively, and the values marginally changed after the blend formation. For the A5F5C2 composite, two overlapping melting endotherms were observed. The *T*_m_ of PA6 shifted to 215.2 °C, which was associated with the melting of the γ-form crystals. The melting of the α-form PA6 crystal occurred at the temperature close to that of the A5F5 blend. For CNT-loaded composites, the *T*_m_ of PA6 was similar to that of the A5F5 blend. The presence of CNT facilitated the growth of the original α-form PA6 crystals. In the blend and composites, PVDF showed overlapped peaks, which might mainly be due to the melting of the original grown and heating-annealed crystals [17]. The melting behavior of PA6 and PVDF in the different samples at a faster precooling rate are depicted in Figure 5b. Compared with the slower rate-cooled corresponding samples, evident multiple melting phenomena were observed for both PA6 and PVDF in the blend/composites. These results are due to the occurrence of more γ-form PA6 crystals (XRD data) and less-stable PVDF crystals at a faster cooling rate. The apparent *T*_m_ values of PA6 and PVDF in different samples are summarized in Table 3. Figure 5c,d show the melting curves of PA6 in A5F5 and A5F5C2 after isothermal crystallization at different *T*_c_s. As anticipated, the *T*_m_ of PA6 increased with increasing *T*_c_ for individual samples. The Hoffman–Weeks (*T*_m_ vs. *T*_c_) plots [38] for individual samples are shown in Figure 5e. A5F5C2 exhibited lower *T*_m_ values of PA6 compared with the other samples at each *T*_c_ because of the 30B-induced formation of γ-form crystals. The equilibrium melting temperature of PA6 (*T*_m_^o^) in different samples was obtained from the plots (intercepts with the *T*_m_ = *T*_c_ straight line). The *T*_m_^o^ of A5F5 (229.6 °C) was close to that of neat PA6 (230.6 °C). The *T*_m_^o^ of the composite with sole 30B inclusion (A5F5C2) was lower than that of A5F5C1T1. This outcome was ascribed to the reduced γ-form crystals because of the presence of CNT. The reason for the low *T*_m_^o^ value of A5F5T2 was not clear. Further experiments are required to elucidate this result.

### 3.4. Mechanical Properties

Typical tensile stress–strain curves of the neat components and their blend/composites are shown in Figure 6a. PA6 showed superior tensile properties to PVDF. The elongation at break (EB) of the blend was lower than those of the neat components due to the immiscible nature between PA6 and PVDF. The composites showed even lower EB than the blend due to the rigid nature of 30B and CNT. Figure 6b shows the comparison of the YS values of the tested samples. Neat PVDF (50 MPa) exhibited lower YS than neat PA6 (85 MPa). The YS (53 MPa) of the blend was within those of the two parent components (close to the PVDF value). The sole addition of CNT into the blend slightly increased the YS (a 13% increase compared to that of the blend), which was higher than the YS of the sole 30B-added composite (a 9% increase). This phenomenon should be associated with the higher rigidity of CNT and the formation of the quasi-co-continuous morphology with the CNT inclusion. YS also increased after the simultaneous addition of 30B and CNT, but the enhancement was not as evident as that of sole 30B- or CNT-added composites.

Figure 6c shows the comparison of the YM of the tested samples. PA6 showed higher YM (2150 MPa) than PVDF (1770 MPa). The blend showed a considerably lower YM (1890 MPa) than that (1960 MPa) calculated by the additivity rule because of interfacial interaction and changes in the phase morphology [39]. YM evidently increased after the formation of the composites. The 30B-loaded composite (A5F5C2) exhibited higher YM than the CNT-loaded composite (A5F5T2). Furthermore, the simultaneous addition of 30B and CNT (A5F5C1T1) showed a value within those of the sole addition of 30B or CNT. The A5F5C2 increased YM approximately by 21% compared with that of the blend. This result could be attributed to the combined effects of PA6-PVDF interface modification by 30B and fine dispersion status of 30B in the blend matrix. Similar to the formulation-dependent trend of YM, the FM results are shown in Figure 6d. The A5F5C2 composite again possessed the highest value among all samples. The FM of A5F5C2 increased by 20% compared to that of the blend.

### 3.5. Rheological Properties

The processability and internal structure change in the immiscible blends after the formation of the nanocomposites can be disclosed from the rheological property measurements. Figure 7a shows the complex viscosity (*η**) as a function of sweep frequency (*ω*) for the samples. PA6 showed Newtonian fluid behavior at almost all frequencies. By contrast, PVDF exhibited Newtonian fluid behavior in the low-frequency region and non-Newtonian fluid behavior at higher frequencies (shear-thinning). PVDF possessed higher *η** than PA6 at all tested frequencies, indicating that PA6 showed a more temperature-sensitive processability than PVDF. The blend showed *η** value in between parent components at all frequencies. The sole 30B-added composite had lower *η** values than the blend. This result could be attributed to the fact that 30B induced more uniform and smaller PVDF domain size. The layered structure of 30B might also play a certain role for the lower *η** values. However, the CNT-loaded composites (cf. A5F5T2 and A5F5C1T1) exhibited evidently higher *η** than the blend as well as non-Newtonian fluid behavior at all frequencies. The shear-thinning behavior started at low frequencies, which suggests that a pseudo-network structure (liquid-like to solid-like transition) was achieved in the samples with the quasi-co-continuous morphology and fine dispersion of 30B/CNT nanofillers. The sweep frequency dependence of storage modulus (*G’*) of the samples is depicted in Figure 7b. *G’* increased with increasing frequency for all samples, and PVDF showed higher *G’* than PA6 at all frequencies. The *G’* values were higher for the blend than its parent components at low frequencies (slightly solid-like behavior due to the non-uniform distribution of the PVDF domain size). The formulation-dependent increase in *G’* after the formation of the composites followed similar trend of the *η** values. The CNT-added composites showed evident increase in *G’* at *ω* ˂ 1 rad/s, and the flattened slopes at low frequencies suggested a solid-like behavior. The pseudo-network structure (development of double percolation) might have led to the observed rheological properties of the CNT-added composites, corresponding to the morphological results. The results revealed that the CNT-added composites had higher melt elasticity than the blend, while the addition of 30B decreased the melt elasticity of the blend.

## 4. Conclusions

Immiscible PA6/PVDF blend and blend-based nanocomposites were successfully fabricated using a twin screw extruder through individual and simultaneous incorporation of 30B and CNT nanofillers. Morphological observations revealed that the presence of 30B reduced the dispersed domain size of PVDF in the blend. Addition of CNT led to the development of a quasi-co-continuous PA6/PVDF morphology. TEM analysis confirmed that 30B and CNT were mostly located in the PA6 matrix phase. DSC data revealed that CNT facilitated the nucleation of PA6 upon crystallization, whereas the addition of 30B barely changed the crystallization of PA6. The crystal structure of PA6 in the sole 30B-added composite was transformed from the α-form to γ-form, as shown by the XRD results. The mechanical properties of A5F5 blend decreased compared to the neat polymers due to the immiscible characteristics. The rigidity of the PA6/PVDF blend considerably increased after the formation of composites. The YS values also increased with increasing CNT and/or the addition of 30B, which enhanced the rigidity of the composites more efficiently than CNT. The measured rheological properties confirmed the formation of a pseudo-network structure in the CNT-loaded composites, and higher CNT loading increased the complex viscosity and *G′* to a higher extent.

## Figures and Tables

**Figure 1 polymers-12-00184-f001:**
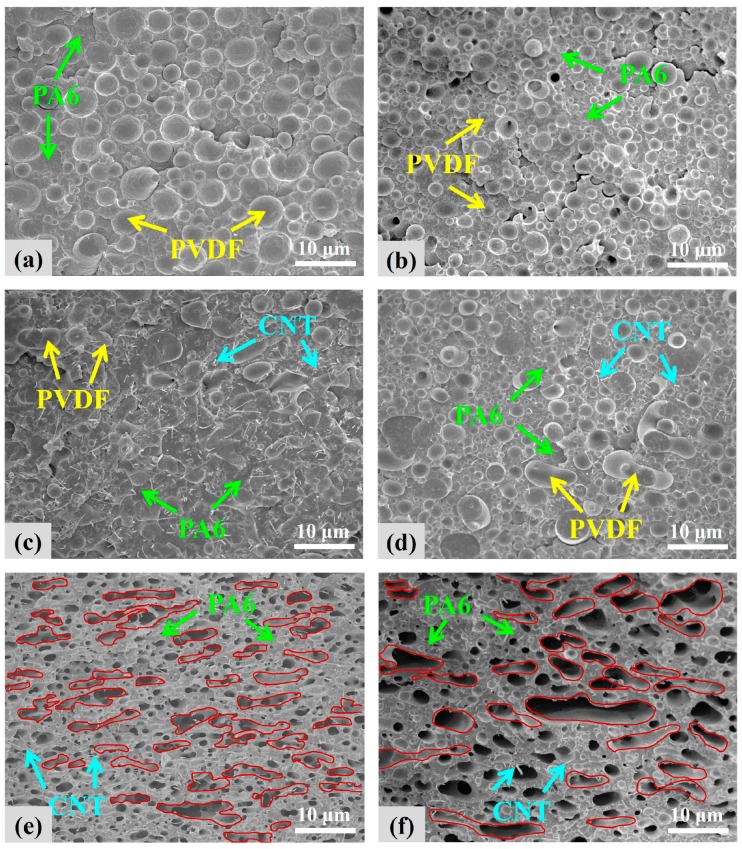
FESEM images of the samples: (**a**) A5F5, (**b**) A5F5C2, (**c**) A5F5T2, (**d**) A5F5C1T1, and PVDF-etched (**e**) A5F5T2, and (**f**) A5F5C1T1.

**Figure 2 polymers-12-00184-f002:**
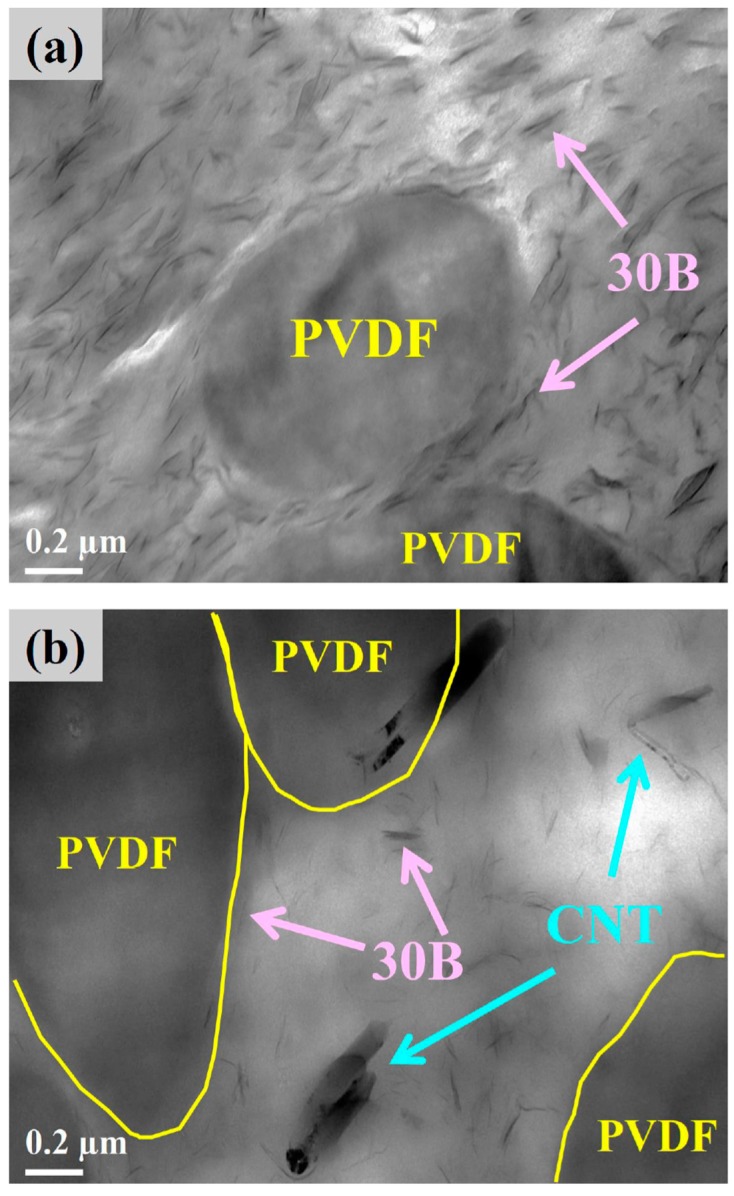
TEM images of (**a**) A5F5C2 and (**b**) A5F5C1T1.

**Figure 3 polymers-12-00184-f003:**
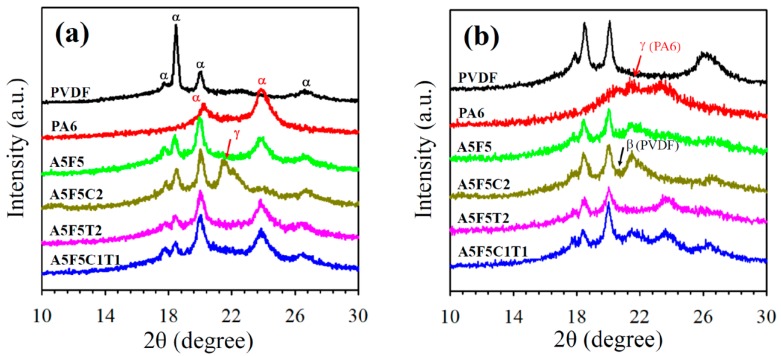
XRD patterns of selected samples prepared through different cooling rates: (**a**) 10 °C/min and (**b**) air quenching.

**Figure 4 polymers-12-00184-f004:**
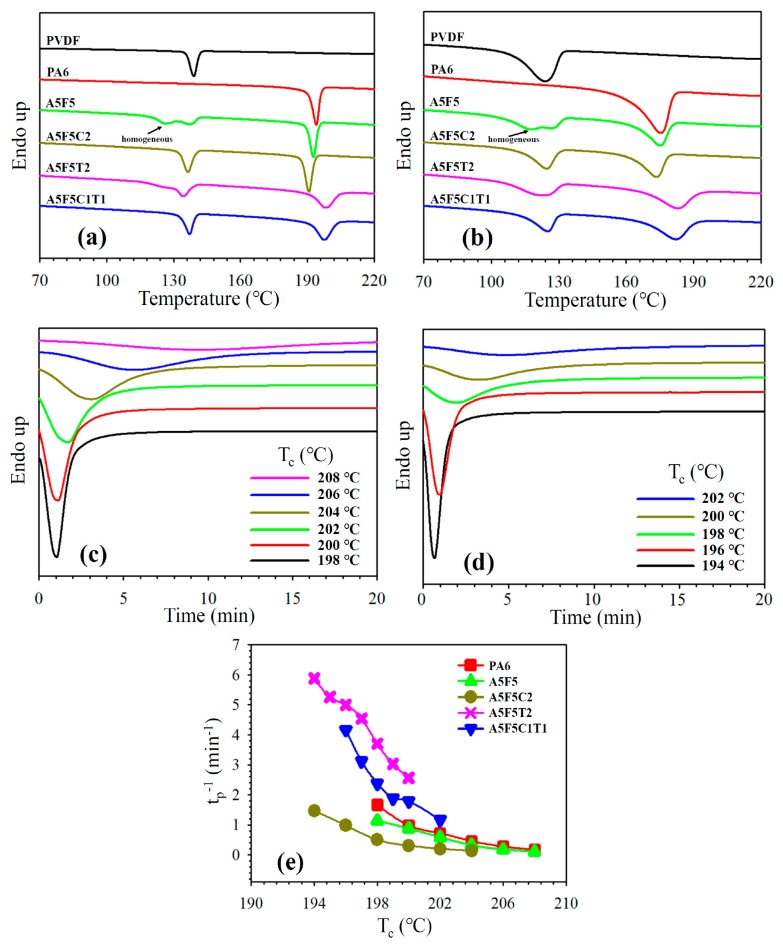
DSC curves of different samples at: (**a**) 5 °C/min cooling and (**b**) 40 °C/min cooling; DSC curves of PA6 isothermally crystallized at indicated *T*_c_ in different samples: (**c**) A5F5 and (**d**) A5F5C2; (**e**) *t*_p_^−1^ of PA6 as a function of *T*_c_ for different samples.

**Figure 5 polymers-12-00184-f005:**
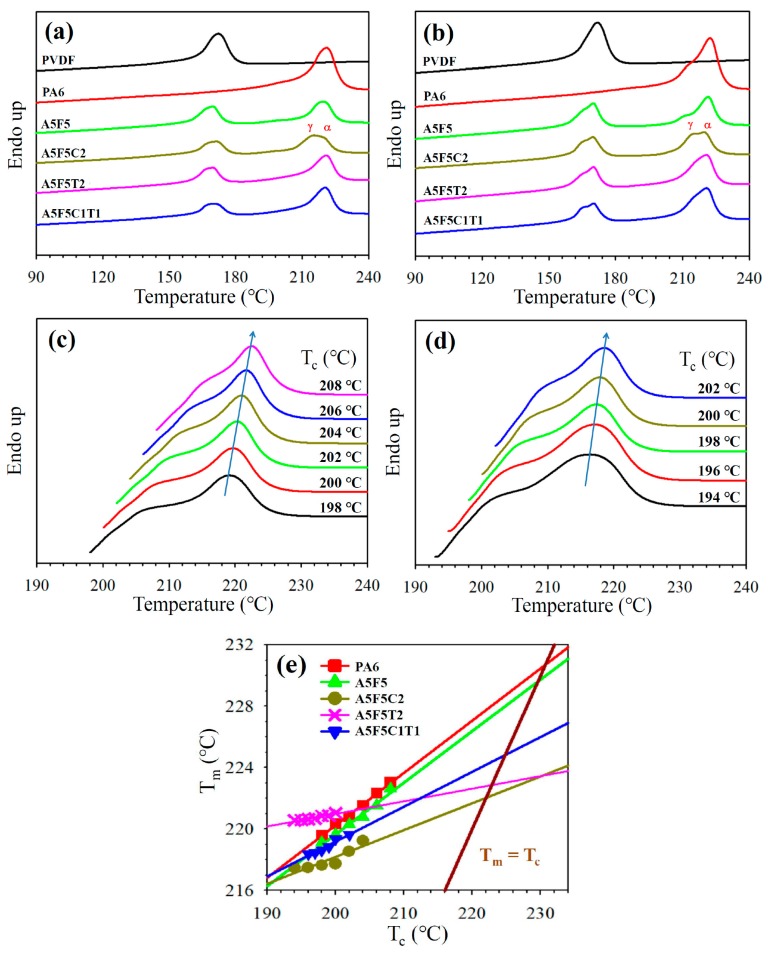
DSC heating curves of different samples: (**a**) 5 °C/min pre-cooled and (**b**) 40 °C/min pre-cooled; DSC melting curves of PA6 isothermally crystallized at indicated *T*_c_ in: (**c**) A5F5 and (**d**) A5F5C2; (**e**) Hoffman-Weeks plots for determining *T*_m_^o^ of PA6 in different samples.

**Figure 6 polymers-12-00184-f006:**
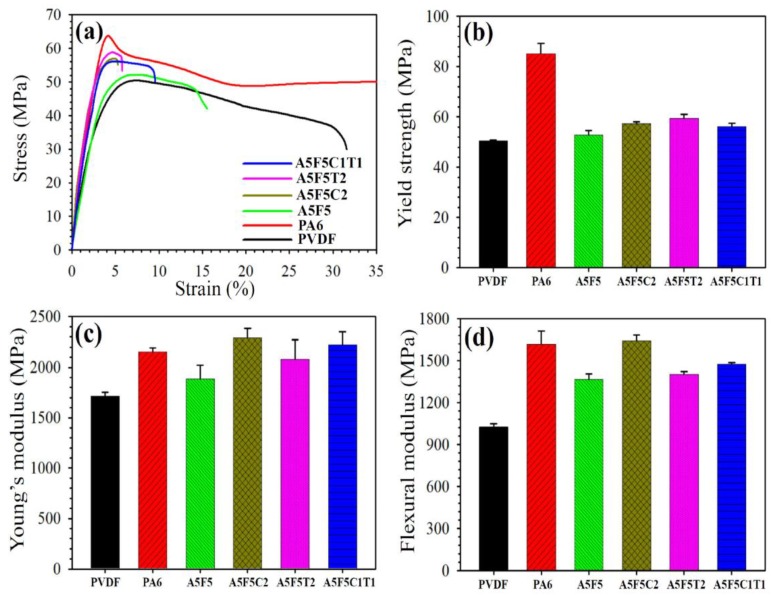
Mechanical properties of PA6, PVDF, blend and composites: (**a**) stress-strain curves, (**b**) yield strength, (**c**) Young’s modulus, and (**d**) flexural modulus.

**Figure 7 polymers-12-00184-f007:**
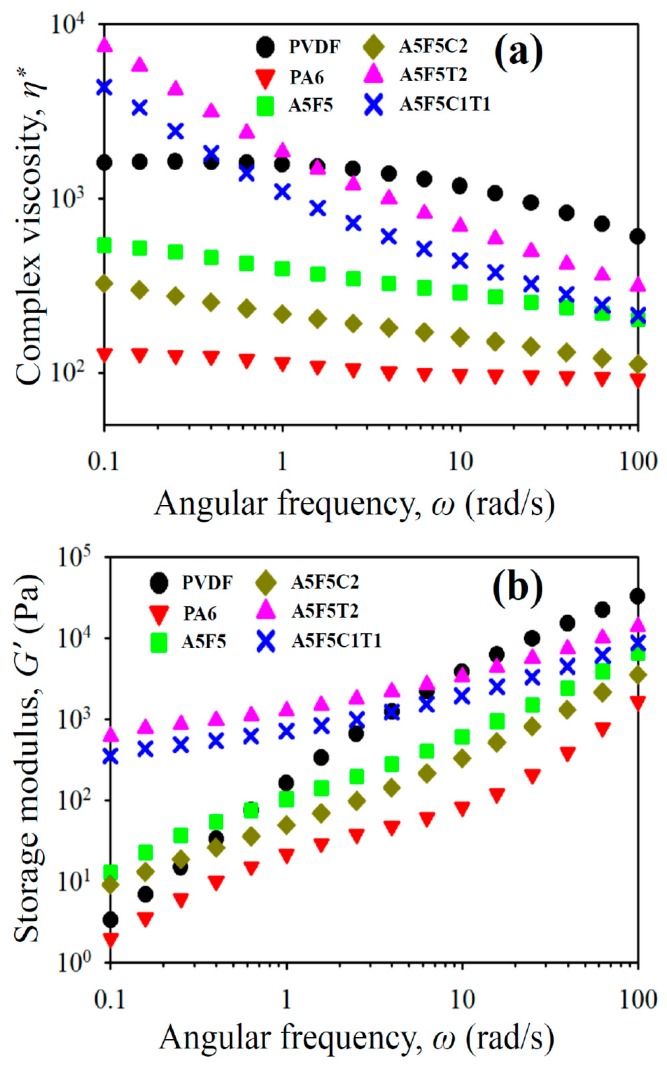
Rheological properties of the samples: (**a**) *η** vs. *ω* and (**b**) *G’* vs. *ω*.

**Table 1 polymers-12-00184-t001:** Surface free energy of studied polymers and nanofillers at 240 °C.

Material	Total Surface Free Energy	Temperature Coefficient	Dispersive Part	Polar Part
	γ (mJm^−2^)	−dγdT^−1^ (mJm^−2^*K)	$\gamma^{d}\;({\mathrm{mJm}}^{-2})$	$\gamma^{p}\;({\mathrm{mJm}}^{-2})$
PVDF [27]	30.3	0	23.3	7.0
PA6 [28]	37.7	0.065 [26]	27.1	10.6
30B [29]	36.0	0.1 [26]	23.0	13.0
CNT [30]	45.3	0	18.4	26.9

**Table 2 polymers-12-00184-t002:** Interfacial tensions and wetting coefficients according to the geometric mean and harmonic mean equations.

Material	Interfacial Tension $\gamma_{12}\;$(mMm^−1^)		Wetting Coefficient, $\omega_{\alpha }$		Location Prediction
	Harmonic Mean Equation	Geometric Mean Equation	Harmonic Mean Equation	Geometric Mean Equation	
$\gamma_{\mathrm{PVDF}-\mathrm{PA}6}$	1.0	0.5			
$\gamma_{\mathrm{PVDF}-30B\;}$	1.8	0.9			
$\gamma_{\mathrm{PA}6-30B}$	0.6	0.3			
$\gamma_{\mathrm{PVDF}-\mathrm{PA}6-30B}$			1.2	1.2	PA6
$\gamma_{\mathrm{PVDF}-\mathrm{CNT}\;}$	12.3	6.7			
$\gamma_{\mathrm{PA}6-\mathrm{CNT}}$	8.8	4.6			
$\gamma_{\mathrm{PVDF}-\mathrm{PA}6-\mathrm{CNT}}$			3.4	4.2	PA6

**Table 3 polymers-12-00184-t003:** Representative DSC thermal data of the samples.

Samples	Properties					
	*T*_cPVDF_ (°C) ^a^	*T*_cPA6_ (°C) ^a^	*χ*_cPVDF_ (%) ^a^	*χ*_cPA6_ (%) ^a^	*T*_mPVDF_ (°C) ^b^	*T*_mPA6_ (°C) ^b^
PVDF	139.0	--	46.2	--	172.0	--
PA6	--	194.1	--	27.5	--	220.9
A5F5	126.1/137.1	192.8	50.4	26.3	169.9	219.4
A5F5C2	136.3	190.8	42.6	27.0	170.8	215.2
A5F5T2	125.3/133.9	197.8	45.9	28.3	169.9	220.7
A5F5C1T1	137.1	197.2	37.6	28.8	170.2	220.4

^a^ 5 °C/min-cooled; ^b^ 20 °C/min-heating after 5 °C/min-cooled.

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
