# Peer review of "Polyamide 6/Poly(vinylidene fluoride) Blend-Based Nanocomposites with Enhanced Rigidity: Selective Localization of Carbon Nanotube and Organoclay"

_polymers, 2020, doi:10.3390/polym12010184_

Round 1

Reviewer 1 Report

In this paper multicomponent materials based on polymer blends formed by PA6/PVDF filled with one or two fillers were prepared. The fillers used in this work were an organoclay and carbon nanotubes. Morphological, mechanical and rheological properties were measured.

Please find below my comments and suggestions:

ABSTRACT:

The description of the work and the findings are clearly stated.

Line 17 “…showed that adding 30B reduced the dispersed domain size of PVDF in the blend”

Adding the word “organomontomorillonite 30B” would clarify the sentence.

INTRODUCTION:

The state of the art is correct, but the structure of the introduction can be improved.

In Paragraph 1, authors emphasize the positive effects of the combination of polymer blends. However it is not sufficiently clear why polyamide and Poly(vinylidene fluoride) (PVDF) are chosen to prepare these multicomponent materials. At the end of this paragraph, the last sentence says “Accordingly, PA6/PVDF blends may integrate the advantages of individual components to present broad applications in various fields”. Could you give some example/s of the potential applications or uses of PA/PVDF polymer blends?

Paragraph 2 is too long. Although paragraph 2 is focused on the effect exerted by fillers in the properties of polymers or polymer blends, in my opinion, paragraph 2 should be revised and re-organized in two/three paragrahs. For example, some ideas might be, one paragraph dedicated to the effects of fillers in polymer crystallization; another on the effect of thermal properties or thermal stability and on mechanical properties.

Considering the structure of this introduction, I consider that the state of the art of this work is correctly attained

2. MATERIALS AND METHODS

Please kindly check the origin of all materials and the technical information of all equipment used in this work.

Also, there are some missing data regarding the preparation of materials for example conditions used to cut the samples with the ultra-cryomicrotome (lines 104-105 : “Ultrathin cryo-microtomed sections of the composites (ca. <100 nm) were prepared for TEM observations”).

In Lines 116-117, Regarding mechanical properties, what was the size of the specimens? How many specimens were tested per sample? Did you follow any standard? ASTM or UNE-ISO? Why did you choose these conditions? Did you do tensile tests?

Similar comments can be said for rheological properties (Lines 117-118), please provide some explanation on why these conditions were selected.

3. RESULTS AND DISCUSSION

In lines 126-127, “The dispersed PVDF exhibited an average domain size of 3.1 μm and was composed of 17% with size <2 μm, 64% with size 2–4 μm, and 20% with size >4 μm.” How was done particle size analysis? Did you use any software?

In Line 268-269, “The reason for the low Tm o value of A5F5T2 was not clear. Further experiments are required to elucidate this result.” Could you provide a tentative explanation for this observation? Was the experiment reproducible?

In Line 290, “The blend showed a considerably lower YM (1886 MPa) than that (1963 MPa) calculated by the additivity rule because of the immiscible characteristic.” Please consider the following reference for discussion of the observed variations in the yield modulus “Polymer Vol. 39 No. 10, pp. 1779-1785, 1998” The effect of interfacial adhesion is discussed here.

4. CONCLUSIONS

Conclusions are clear and concise. The main findings of this research are clearly indicated, although some points are omitted such as the drop in mechanical properties of the sample A5F5. For this, I suggest authors to reconsider revising the main conclusions of the document.

In my opinion the research carried out in this work is good, based on the abovementioned, I suggest the paper is acceptable for publication after minor revisions.

Author Response

Response to Reviewer 1 Comments

The reviewers’ comments and helpful suggestions on the manuscript (Polymers-679322) are sincerely appreciated. The followings are the point-by-point response to each comment/suggestion. The changes have been highlighted in the revised manuscript with red color.

Reviewer 1:

Comments and Suggestions for Authors

In this paper multicomponent materials based on polymer blends formed by PA6/PVDF filled with one or two fillers were prepared. The fillers used in this work were an organoclay and carbon nanotubes. Morphological, mechanical and rheological properties were measured.

Please find below my comments and suggestions:

ABSTRACT:

The description of the work and the findings are clearly stated.

Line 17 “…showed that adding 30B reduced the dispersed domain size of PVDF in the blend”

Adding the word “organomontomorillonite 30B” would clarify the sentence.

Response: Thank you for the suggestion. In the Abstract, “organomontomorillonite 30B” was used instead of “30B”. The Experimental part added organomontomorillonite denoted as 30B, and it was done in the revised manuscript.

INTRODUCTION:

The state of the art is correct, but the structure of the introduction can be improved.

In Paragraph 1, authors emphasize the positive effects of the combination of polymer blends. However it is not sufficiently clear why polyamide and poly(vinylidene fluoride) (PVDF) are chosen to prepare these multicomponent materials. At the end of this paragraph, the last sentence says “Accordingly, PA6/PVDF blends may integrate the advantages of individual components to present broad applications in various fields”. Could you give some example/s of the potential applications or uses of PA/PVDF polymer blends?

Response: The suggestion is appreciated and helpful. The following sentences are added into Paragraph 1 “The polar functional groups (amide) of PA6 tend to absorb moisture, resulting in poor stability after prolonged exposure to air environment. Poly(vinylidene fluoride) possesses good hydrolysis resistance and weatherability. Through polymer blending approach, it is expected to overcome the drawbacks of parent components [1–3]. The PA6/PVDF blends and related composites with fine dielectric properties can be applied in electronic field such as embedded capacitors [4]. PA6/PVDF blend-based membranes may be used for separating gases such as CO2, N2, CH4’’ [5].

Reference:

Ho, J.C.; Wei, K.H. Induced g → a crystal transformation in blends of polyamide 6 and liquid crystalline copolyester. Macromolecules 2000, 33, 5181-5186. Chiu, F.C. Poly(vinylidene fluoride)/polycarbonate blend-based nanocomposites with enhanced rigidity-selective localization of carbon nanofillers and organoclay. Test. 2017, 62, 115-123. Elnabawy, E.; Hassanain, A.H.; Shehata, N.; Popelka, A.; Nair, R.; Yousef, S.; Kandas, I. Piezoelastic PVDF/TPU nanofibrous composite membrane: fabrication and characterization. Polymers 2019, 11, 1634-1648. Mao, H.; Zhang, T.; Huang, T.; Zhang, N.; Wang; Y.; Yang, J. Fabrication of high-k poly(vinylidene fluoride)/nylon 6/carbon nanotube nanocomposites through selective localization of carbon nanotubes in blends. Int. 2017, 66, 604-611. Duarte, J.; Cherubini, C.C.; Snatos, V.D.; Schenider, A. Zeni, M. Poly(vinylidene fluoride) (PVDF) and nylon 66 (PA66) membranes applied the process of gas separation. Procedia Eng. 2012, 44, 1146-1149

Paragraph 2 is too long. Although paragraph 2 is focused on the effect exerted by fillers in the properties of polymers or polymer blends, in my opinion, paragraph 2 should be revised and re-organized in two/three paragrahs. For example, some ideas might be, one paragraph dedicated to the effects of fillers in polymer crystallization; another on the effect of thermal properties or thermal stability and on mechanical properties.

Response: The original paragraph 2 is separated into two parts in the revised manuscript according to the reviewer’s suggestion.

Considering the structure of this introduction, I consider that the state of the art of this work is correctly attained

MATERIALS AND METHODS

Please kindly check the origin of all materials and the technical information of all equipment used in this work.

Also, there are some missing data regarding the preparation of materials for example conditions used to cut the samples with the ultra-cryomicrotome (lines 104-105 : “Ultrathin cryo-microtomed sections of the composites (ca. <100 nm) were prepared for TEM observations”).

Response:  The origin of all materials and the technical information of all equipments used in this work were carefully checked (without mistake). The TEM samples (ca. <100 nm) were prepared by ultra-cryomicrotoming at -130 °C. This point was addressed in the Experimental part.

In Lines 116-117, Regarding mechanical properties, what was the size of the specimens? How many specimens were tested per sample? Did you follow any standard? ASTM or UNE-ISO? Why did you choose these conditions? Did you do tensile tests?

Response: We did the tensile tests for the prepared samples. For yield strength and Young’s modulus measurement, dumbbell-shaped specimen was used according to ASTM D638. For flexural modulus measurement, cuboid-shaped specimen was used (L : W : H = 62.5 : 12.5 : 3.2 mm) according to ADTM D790.  Six specimens were tested for each formulation, and the average value was reported. The conditions we chose for the different tests were to obtain the proper data for comparison. These were addressed in the revised manuscript.

Similar comments can be said for rheological properties (Lines 117-118), please provide some explanation on why these conditions were selected.

Response: The authors followed the commonly used conditions for the rheological properties measurement according to the previously published articles [2].

Chiu, F.C. Poly(vinylidene fluoride)/polycarbonate blend-based nanocomposites with enhanced rigidity-selective localization of carbon nanofillers and organoclay. Polym. Test. 2017, 62, 115-123. RESULTS AND DISCUSSION

In lines 126-127, “The dispersed PVDF exhibited an average domain size of 3.1 μm and was composed of 17% with size <2 μm, 64% with size 2–4 μm, and 20% with size >4 μm.” How was done particle size analysis? Did you use any software?

Response: The SG capture software was used to measure the domain size of PVDF from SEM images. This point was addressed in the revised manuscript.

In Line 268-269, “The reason for the low Tm value of A5F5T2 was not clear. Further experiments are required to elucidate this result.” Could you provide a tentative explanation for this observation? Was the experiment reproducible?

Response:  The result is reproducible. The authors conducted this experiment three times, and then obtained the same result.

In Line 290, “The blend showed a considerably lower YM (1886 MPa) than that (1963 MPa) calculated by the additivity rule because of the immiscible characteristic.” Please consider the following reference for discussion of the observed variations in the yield modulus “Polymer Vol. 39 No. 10, pp. 1779-1785, 1998” The effect of interfacial adhesion is discussed here.

Response:  As per the reviewer’s suggestion, the statement of “because of interfacial interaction and changes in the phase morphology [6]” was added for the discussion of YM data observation.

Liu, Z.H.; Maréchal, P.; Jérôme, R. Blends of poly(vinylidene fluoride) with polyamide 6: interfacial adhesion, morphology and mechanical properties. Polymer 1998, 39, 1779-1785. CONCLUSIONS

Conclusions are clear and concise. The main findings of this research are clearly indicated, although some points are omitted such as the drop in mechanical properties of the sample A5F5. For this, I suggest authors to reconsider revising the main conclusions of the document.

Response:  The statement of “The mechanical properties of A5F5 blend decreased compared to the neat polymers because of the immiscible characteristics.” was added in the revised manuscript.

Reviewer 2 Report

The paper entitled “Polyamide 6/poly(vinylidene fluoride) blend-based nanocomposites with enhanced rigidity: Selective localization of carbon nanotube and organoclay” presents an overall interesting investigation of an immiscible blend formation, and the impact of different nanofillers on their properties, both when added independently and also when added concomitantly. The paper is well written and detailed, with a strong bibliography supporting the hypothesis and the statements and a good discussion of the obtained results. The reviewer just observed a few points that should be improved prior publication, to increase the standards of the paper.

Page 1 line 33: change academe into Academia.

In the introduction paragraph, when discussing prior publications about blend nanocomposites, when Authors switch from the discussion of the PVDF/PCL to PA6/PVDF blend the switch is not clear. Authors should rephrase that part to improve readability. (Page 2 lines 61-65).

Page 2, line 65: Authors state that co-continuous morphology is promote by “finer” interaction of CNT and PA6: what do the Authors mean with finer interaction? Please discuss this effect in a more appropriate way.

In the paragraph 3.1. Phase morphology and selective localization of nanofillers Authors should first introduce the theoretical evaluation of nanofillers miscibility and location in the two polymeric matrices, that is presently appearing in the concluding section of the paragraph. The experimental validation of such data should appear subsequently to substantiate the theoretical hypothesis. Moreover, in this paragraph Authors state that A5F5C1T1 sample shows a co-continuous morphology. However, to the reviewer, seems that A5F5T2 has a higher evidence of such co-continuos morphology, also based on the etched SEM micrograph where thinner and more interconnected holes are evident. Please comment and amend the discussion accordingly.

In Figure 2 it would be beneficial the comparison of the third nanocomposite blend that I presently missing (A5F5T2). Authors should add it to the Figure and comment it in the main text too.

Page 8 line 228: 40°C/min

Both in Figures 4 and 5 a reference for the Y axis should be provided, while it is correct that no numbering appears in the Y axis, at least scale should be added, and an inset with a scale bar with a reference unit should be inserted in each DSC graph.

Please round the YM values to at least the tens of units.

Author Response

Response to Reviewer 2 Comments

In my opinion the research carried out in this work is good, based on the abovementioned, I suggest the paper is acceptable for publication after minor revisions.

Reviewer 2:

Comments and Suggestions for Authors

The paper entitled “Polyamide 6/poly(vinylidene fluoride) blend-based nanocomposites with enhanced rigidity: Selective localization of carbon nanotube and organoclay” presents an overall interesting investigation of an immiscible blend formation, and the impact of different nanofillers on their properties, both when added independently and also when added concomitantly. The paper is well written and detailed, with a strong bibliography supporting the hypothesis and the statements and a good discussion of the obtained results. The reviewer just observed a few points that should be improved prior publication, to increase the standards of the paper.

Page 1 line 33: change academe into Academia.

Response: Thanks for the suggestion. The suggested change (Academia) was done in the revised manuscript.

In the introduction paragraph, when discussing prior publications about blend nanocomposites, when Authors switch from the discussion of the PVDF/PCL to PA6/PVDF blend the switch is not clear. Authors should rephrase that part to improve readability. (Page 2 lines 61-65).

Response:  In this introduction paragraph, authors discussed different previously reported blend-based nanocomposites. The rephrase has been done accordingly in the revised manuscript.

Page 2, line 65: Authors state that co-continuous morphology is promote by “finer” interaction of CNT and PA6: what do the Authors mean with finer interaction? Please discuss this effect in a more appropriate way.

Response: The authors mentioned the following statements: “Phase transformation from sea-island to co-continuous by CNT loading which increased the melt viscosity of the PA6 phase. “

In the paragraph 3.1. Phase morphology and selective localization of nanofillers Authors should first introduce the theoretical evaluation of nanofillers miscibility and location in the two polymeric matrices, that is presently appearing in the concluding section of the paragraph. The experimental validation of such data should appear subsequently to substantiate the theoretical hypothesis. Moreover, in this paragraph Authors state that A5F5C1T1 sample shows a co-continuous morphology. However, to the reviewer, seems that A5F5T2 has a higher evidence of such co-continuos morphology, also based on the etched SEM micrograph where thinner and more interconnected holes are evident. Please comment and amend the discussion accordingly.

Response:  The authors have added the following statement in the manuscript. “A5F5T2 composite showed evident quasi-co-continuous morphology.” And we think the original arrangement of this paragraph is fine (might not need to amend it). We will consider the reviewer’s suggestion in our following articles.

In Figure 2 it would be beneficial the comparison of the third nanocomposite blend that I presently missing (A5F5T2). Authors should add it to the Figure and comment it in the main text too.

Response: The suggestion is appreciated. As the authors mentioned, only representative samples are shown in Figure 2. From our viewpoint, A5F5T1 image is good enough to identify the localization of CNT within PA6 phase. A5F5T2 shows similar CNT localization in the blend (not shown for brevity).

Page 8 line 228: 40 °C/min

Response: Thanks for the suggestion. The correction has been done accordingly in the revised manuscript.

Both in Figures 4 and 5 a reference for the Y axis should be provided, while it is correct that no numbering appears in the Y axis, at least scale should be added, and an inset with a scale bar with a reference unit should be inserted in each DSC graph.

Response: Generally, Y-axis (Endo up) scale bar is not required in the DSC graph as reported in most of the DSC data-included papers.

Please round the YM values to at least the tens of units.

Response: The authors round the YM values to the tens of units in the revised manuscript according to the reviewer’s suggestion.

Reviewer 3 Report

This manuscript prepared the immiscible Polyamide 6 (PA6)/poly(vinylidene fluoride) (PVDF) blend-based nanocomposites inducing carbon nanotube (CNT) and organo-montmorillonite (30B) as reinforcing nanofillers. The TEM results revealed that both fillers were mainly located in the PA6 matrix phase. As a result, the addition of 30B can reduce the dispersed domain size of PVDF, facilitate the formation of g-form PA6 crystals increased the Young’s and flexural moduli, and the addition of CNT can help the formation of a quasi-co-continuous (pseudo-network) PA6-PVDF morphology, increased crystallization temperature of PA6. However, there are several issues the authors need to address before considering for publishing.

1, Figure 1, how to confirm the marked section in images is the CNT or 30B, please add the corresponding measurement of pure CNT and 30B. Is the DMF only reacted with PVDF for etching, no effect on PA6, CNT or 30B?

2, Are the SEM images measured cross-section? If not, how to prepare the samples? The same in Figure 2, how to make the composites films sample in TEM measurement?

3, TEM results revealed that both fillers were mainly located in the PA6 matrix phase, how the 30B in PA6 reduce the dispersed domain size of PVDF? And the CNT in PA6 effect the PA6-PVDF composites morphology.

Author Response

Response to Reviewer 3 Comments

Reviewer 3:

Comments and Suggestions for Authors

This manuscript prepared the immiscible Polyamide 6 (PA6)/poly(vinylidene fluoride) (PVDF) blend-based nanocomposites inducing carbon nanotube (CNT) and organo-montmorillonite (30B) as reinforcing nanofillers. The TEM results revealed that both fillers were mainly located in the PA6 matrix phase. As a result, the addition of 30B can reduce the dispersed domain size of PVDF, facilitate the formation of g-form PA6 crystals increased the Young’s and flexural moduli, and the addition of CNT can help the formation of a quasi-co-continuous (pseudo-network) PA6-PVDF morphology, increased crystallization temperature of PA6. However, there are several issues the authors need to address before considering for publishing.

1, Figure 1, how to confirm the marked section in images is the CNT or 30B, please add the corresponding measurement of pure CNT and 30B. Is the DMF only reacted with PVDF for etching, no effect on PA6, CNT or 30B?

Response: In Figure 1 the authors marked only CNT location, and the size of 30B was too small to be observed by SEM. For the location of 30B and CNT, authors used TEM for the further observation as illustrated in Fig.2. DMF only reacts with the PVDF phase as reported.

2, Are the SEM images measured cross-section? If not, how to prepare the samples? The same in Figure 2, how to make the composites film sample in TEM measurement?

Response: SEM images were observed from cross-section of (liquid nitrogen) cryo-fractured specimens. Regarding TEM samples preparation, the ultra-cryomicrotome technique was employed at -130 °C. These points are addressed in the Experimental part.

3, TEM results revealed that both fillers were mainly located in the PA6 matrix phase, how the 30B in PA6 reduce the dispersed domain size of PVDF? And the CNT in PA6 effect the PA6-PVDF composites morphology.

Response: 30B was also detected at the interface between PA6 and PVDF, and then modified the interfacial tension in reducing the dispersed domain size of PVDF. CNT was found in the PA6 phase and also observed in the interfacial region of PA6-PVDF phases. The viscosity alteration of PA6 with the filler inclusion also played a role in the morphology evolution of the blends.
